# Identification of a *TPP1* Q278X Mutation in an Iranian Patient with Neuronal Ceroid Lipofuscinosis 2: Literature Review and Mutations Update

**DOI:** 10.3390/jcm11216415

**Published:** 2022-10-29

**Authors:** Tayebeh Baranzehi, Dor Mohammad Kordi-Tamandani, Maryam Najafi, Ali Khajeh, Miriam Schmidts

**Affiliations:** 1Departement of Biology, University of Sistan and Baluchestan, Zahedan 9816745845, Iran; 2Pediatric Genetics Division, Center for Pediatrics and Adolescent Medicine, University Hospital Freiburg, Freiburg University Faculty of Medicine, 79106 Freiburg, Germany; 3Genome Research Division, Human Genetics Department, Radboud University Medical Center, 6525 GA Nijmegen, The Netherlands; 4Department of Pediatric, Children and Adolescent Health Research Center, Zahedan University of Medical Sciences, Zahedan 9816745845, Iran; 5CIBSS—Center for Integrative Biological Signaling Studies, University of Freiburg, 79104 Freiburg, Germany

**Keywords:** CLN2, *TPP1* gene, c.C832T, R278*

## Abstract

Neuronal ceroid lipofuscinoses type 2 (CLN2), the most common form of Batten disease, is caused by TPP1 loss of function, resulting in tripeptidyl peptidase-1 enzyme deficiency and cerebral accumulation of lipopigments. Clinical hallmarks include epileptic seizures, vision loss, progressive movement disorder, ataxia, and eventually death. Diagnosis is often delayed due to the rarity of the conditions. Results: Here, we report a case presenting with clinical features of CLN2, carrying a homozygous novel nonsense variant in *TPP1* (NM_000391:c.C832T, (p.Q278*), rs1352347549). Moreover, we performed a comprehensive literature review regarding previously identified disease-causing TPP1 mutations and genotype-phenotype correlations. Conclusion: Depending on the type of mutation, different phenotypes are observed in patients with CLN2, suggesting that the severity of phenotypes is related to the genotype of the patients.

## 1. Introduction

Neuronal ceroid lipofuscinoses (NCLs) represent a group of rare clinically and genetically heterogeneous progressive neurodegenerative lysosomal storage disorders (LSD), mainly affecting children aged 2–6 years, caused by loss-of-function mutations in CLN-genes. With the exception of CLN4 which is autosomal-dominantly inherited, all other forms follow an autosomal-recessive mode of inheritance [1,2,3]. Children mostly show normal psychomotoric development until progressive development of seizures, vision loss, intellectual and motor decline, cognitive impairment, and eventually premature death. Despite the rarity of this disease, 300–350 cases of NCLs are reported annually in the US, and its global incidence is one per 100,000 live births (orphaned; ORPHA 79264). The highest incidence rates have been reported in some Nordic countries (one per 14,000 in Iceland) [1,4,5], while incidence is unknown for many other countries, including Iran.

The classification of NCLs is based on the defective gene or protein as well as the age of onset of clinical symptoms. To date, more than 537 disease-causing mutations have been published in 13 different genes [6], https://www.ucl.ac.uk/ncl-disease/, accessed on 20 June 2022). Defective genes encode for lysosomal proteins with enzyme function (Palmitoyl-protein thioesterase 1, (CLN1); Tripeptidyl-peptidase 1 (CLN2), Cathepsin D (CLN10), and Cathepsin F (CLN13)); transmembrane lysosomal proteins (Battenin (CLN3) and ATP13A2 (CLN12); soluble lysosomal proteins (CLN5 and CLN11 (Granulin precursor, GRN)); a transmembrane endosomal protein of the major facilitator superfamily domain-containing protein-8 (CLN7); endoplasmic reticulum (ER) transmembrane proteins (CLN6 and CLN8); as well as soluble cytosolic proteins (CLN4 (DNAJC5) and CLN14 (KCTD7)), Figure 1.

According to the age of onset, CLNs are classified into six subtypes, including a congenital form (CLN10, MIM610127), an infantile form (CLN1, Santavuori-Haltia disease; MIM256730), a late infantile form (CLN2, Jansky Bielschowsky disease; MIM204500), variable late infantile forms (CLN5, MIM256731; CLN6, MIM601780; CLN7, MIM610951 and CLN8, MIM600143), a juvenile form (CLN3; Spiel-Meyer-Vogt-Sjogren disease; MIM304200), and an adult form (CLN4; Kufs disease; MIM204300) [7,8,9,10,11,12]. Accumulation of mitochondrial ATP synthase subunit C or Saposin A and D (autofluorescent lipopigments) in the lysosomal storage bodies of different cell types, specifically neurons, are the main neuropathological features in NCLs [13,14].

*CLN2* (OMIM#607998) is located on the short arm of chromosome 11 (11p15), consisting of 13 exons and 12 introns [5,15,16,17,18]. *CLN2* encodes for Tripeptidyl Peptidase 1 (*TPP1*), a lysosomal aminopeptidase cleaving N-terminal tripeptides from small polypeptides. In humans, this gene encodes a 61-kDa inactive precursor (proenzyme) containing 563 amino acids (aa), including a 368-aa catalytic domain, a 19-aa leader sequence (signal peptide), and a 176-aa prodomain. A signal peptide enables ER-specific localization for cotranslational cleavage in the ER lumen. This is followed by N-glycosylation at the Golgi, enabling proper folding, enhancing protein stability, activity, and intracellular enzyme targeting. Transport to the lysosome is enabled by mannose-6-phosphate receptor binding. At the lysosome, CLN2 is first further processed into a 50 kDa polypeptide followed by cleavage into the mature 368 amino acid (48 kDa) enzyme by lysosomal exoglycosidases such as neuraminidases [19,20,21]. Lysosomal degradation pathways involving TPP1 are summarized in Figure 2.

Overlapping clinical features and extensive genetic heterogeneity renders next-generation sequencing (NGS) the most efficient and precise genetic diagnostic method if NCL is suspected. In the present study, we describe the identification of a novel homozygous *TPP1* variant in an Iranian patient with CLN2 and present an overview of previously published CLN2 disease alleles as well as genotype-phenotype correlations.

## 2. Materials and Methods

The research ethics committee ethically approved this study of the Mashhad University of medical sciences committee, and all study participants signed the written informed consent (IR.MUMS.REC.1395.534). Genetic diagnostics was performed under the Radboudumc Innovative Genetics Diagnostics program.

### 2.1. DNA Extraction

Genomic DNA was extracted from the peripheral blood of the proband and his family members using the established salting-out method, and the integrity of the extracted DNA was assessed by gel electrophoresis (1% agarose).

### 2.2. Whole Exome Sequencing

Two microgrammes of DNA of the proband was subjected to exome capture using Agilent SureSelect Human All Exon V6 Kit (Agilent, Sata Clara, CA, USA) and sequencing performed on an Illumina HiSeq 2500 (Illumina, San Diego, CA, USA) sequencer for an average 50× sequencing depth, resulting in sequences of greater than 100 bases from each end of the fragments (Novogene, Cambridge, UK). GATK-based pipeline [22] was used for variant calling using Burrows–Wheeler alignment [23] to perform sequence alignment to the GRCh37/UCSC hg19 reference genome. SNV were detected using VarScan version 2.2.5, MuTec, Iranome, gnomAD, and Greater Middle East (GME) Variome Project databases were used for population-frequency specific variant filtering.

### 2.3. Polymerase Chain Reaction (PCR) and Sanger sequencing

We used 1 µL of genomic DNA for PCR reactions. Furthermore, 50 pmol of each forward (5′-GGGATCACTGTGGAGTCAAAG-3′) and reverse (5′-AGACCTGGCTCAGTTCATGC-3′) primers, 10 µL of 2× PCR Master Mix (Sina Clon Cat. No.: MM2062), and 12 µL of distilled water were used in a final volume of 25 µL. PCR reactions were carried out using an initial incubation step at 95 °C for 5 min followed by 35 cycles at 94 °C for 40 s, annealing at 60 °C for 40 s, and extension at 72 °C for 40 s. A final extension step at 72 °C for 10 min was used. We analyzed 5 µL of PCR products using a 2% agarose gel for electrophoresis. Subsequently, the PCR products were purified and sequenced using Sanger sequencing.

### 2.4. Literature Search

Search terms “TPP1” or “CLN2” were used in PubMed to retrieve relevant human mutation reports and ClinVar and ACMG databases were searched for reported TPP1 alleles.

## 3. Results

### 3.1. Clinical Assessment

A 5-year-old patient diagnosed with epilepsy and ataxia was referred to our clinic after a generalized tonic-clonic seizure. The boy was born to a consanguineous Iranian family (parents were first cousins) with an unremarkable family history for neurological disorders; however, the mother experienced two miscarriages due to unknown reasons at 12 and 14 weeks of gestation (Figure 3A).

The patient was born naturally at term with a birth weight of 3.2 kg. Perinatal course and development were normal. Motor and developmental milestones were reached at normal age, for example, free walking at 12 months and talking in sentences at 24 months. At four years of age, a generalized tonic-clonic seizure occurred followed by rapid cognitive and motoric regression, including losing the ability to walk, loss of speech, progressive ataxia, spasticity, and difficulties swallowing. In the last examination, the patient was found to be legally blind. Hematology tests revealed mild anemia (erythroctyes 4.48/µL, hematocrit 32.6%, and hemoglobin 10.6 g/dL). Serum glutamic-oxaloacetic transaminase and alkaline phosphatase were increased (46 mg/dL and 430 U/L, respectively). Magnetic resonance imaging (MRI) findings revealed only mild supratentorial dilation of the ventricular system. There was cerebellar vermian and significant cerebellar hemispheres hypoplasia with prominent 4th ventricle and cistern magna, thinning of superior cerebellar peduncles and cystic lesion of cerebrospinal fluid (CSF) intensity communication with 4th ventricle, and delayed myelinisation of the periventricular white matter and centrum semiovale (Figure 4). The patient underwent chest physiotherapy and received Levetiracetam, Sodium valproate, Nitrazepam, Tetracosactide, Vitamin B6, and Biotin. At the age of 4 years and 8 months, the patient was admitted to the ICU with a GCS of 4.5 and died after 2 months.

### 3.2. Genetic Analysis

Whole-exome sequencing and analysis revealed a novel homozygous loss of function (LOF) variant in exon 7 of *TPP1* (c.C832T: p.Gln278Ter, rs135234754). The allelic frequency of this variant in the gnomAD database is 0.000003977 while no frequency has been reported in ExAC or the 1000 genomes databases. The homozygous variant co-segregated with the disease in family, with both parents and a healthy sibling found to be heterozygous carriers (Figure 3B–E).

## 4. Discussion

CLN2 is a specific subtype of NCLs caused by TPP1 loss of function, resulting in seizures, vision loss, dementia, cerebellar ataxia, sleep disorders, progressive psychomotor decline, and death in the first decade of life [24]. Early symptoms of CLN2 include seizures and photosensitivity [25]; however, seizures can rarely be absent [3,26]. In line with these findings, our case initially presented with a generalized tonic-clonic seizure. Microcephaly, absent in our case, has additionally been reported in a few cases, including a 9-year-old Australian boy [27] and an 8-year-old boy from Turkey [7], both of whom had non-consanguineous parents. Further common findings include cerebellar atrophy, resulting in ataxia and/or tremor including in our case [7,13,28,29,30,31] (Table 1). Further, vision loss and/or optic atrophy was observed in most patients. Further frequent manifestations include spasticity as well as swallowing and sleeping problems.

Based on PubMed publications and submissions in ClinVar, more than 155 potentially disease-associated *TPP1* variants have been reported in total (Appendix A and Figure 5). TPP1 is required in lysosomes for protein degradation and loss of TPP1 activity results in the accumulation of auto fluorescent lysosomal storage material in various cell types, including neurons [39]. Overall, missense variants are most frequently identified (63, 48%), followed by frameshift (21, 16%), and stop-gain (17, 13%) variants [6,17]. In addition to CLN2, TPP1 dysfunction can also result in autosomal-recessive Spinocerebellar Ataxia 7 (MIM609270) [2,32]. This phenotype is characterized by cerebellar ataxia, tremor, dysarthria, and nystagmus with an age of onset during the first or second life decade. Likely, alleles causing spinocerebellar ataxia result on proteins with some residual function, hence the hypomorphic phenotype in comparison to CLN2 disease alleles with later age of onset and an overall less severe clinical picture [40]. In line with this, missense alleles can be frequently identified in Spinocerebellar Ataxia 7, while approximately 60% of patients with CLN2 carry two common LOF mutations: the splice acceptor site mutation (c.509-1G > C) and the stop-gain mutation in exon 6 (c.622C > T, p.R208X), which may occur homozygously or heterozygously *in trans* (compound-heterozygously) with other disease alleles [1,8,41]. In the present study, we report a novel homozygous pathogenic stop-gain variant (p.Arg278*) in an Iranian family. This allele results in a truncation within the peptidase S_53 domain. Up to now, 73 *TPP1* null variants have been reported in the ClinVar, six of which are found in exon 7.

The role of genetic diagnostics in CLN, especially CLN2, has become more important than ever. In pediatric cases of CLN2, replacement of the dysfunctional TPP1 enzyme with a functional recombinant enzyme (Cerliponase alpha) by intraventricular injection has been reported to effectively delay disease progression and to stabilize the loss of motor and language function [24,41,42]. To achieve the best possible outcomes, an early genetic diagnosis will be essential. For our case, diagnosis came too late for possible interventions.

Further, genetic testing enables prenatal and pre-implantation diagnostics as well as carrier identification, offering novel family planning options.

## Figures and Tables

**Figure 1 jcm-11-06415-f001:**
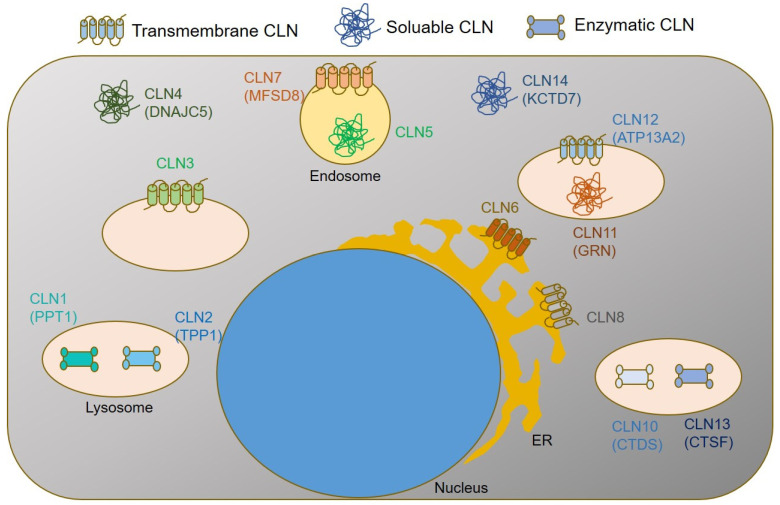
Simplified graphical summary of subcellular CLN protein localizations and functions.

**Figure 2 jcm-11-06415-f002:**
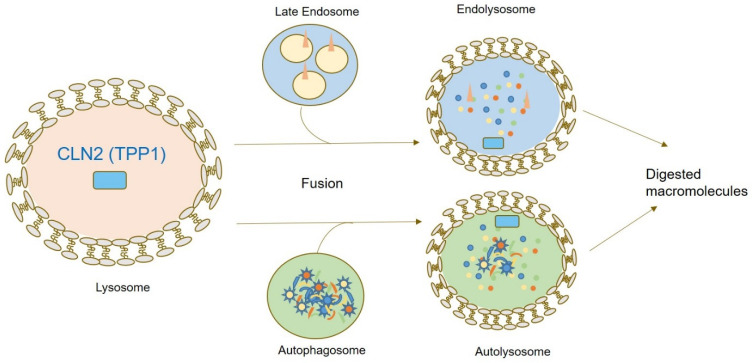
Graphical summary of CLN2/TPP1 localization and function within the cell. TPP1 represents a peptidase contributing to N-terminal protein degradation. Upon fusion of autophagosomes and late endosomal vesicles with lysosomal vesicles, lysosomal enzymes including TPP1 enable digestion of macromolecules.

**Figure 3 jcm-11-06415-f003:**
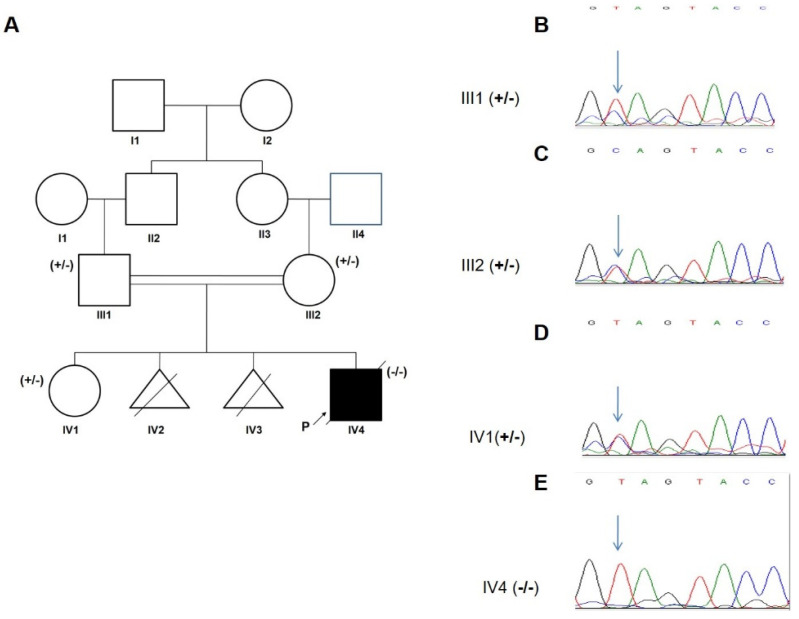
Family pedigree and segregation pattern. (**A**) Pedigree of the family; +/−: heterozygous, −/−: homozygous. (**B**–**E**) Sanger sequencing chromatograms of the proband, his parents and healthy sister showing the TPP1 c. C832T variant indicated by an arrow.

**Figure 4 jcm-11-06415-f004:**
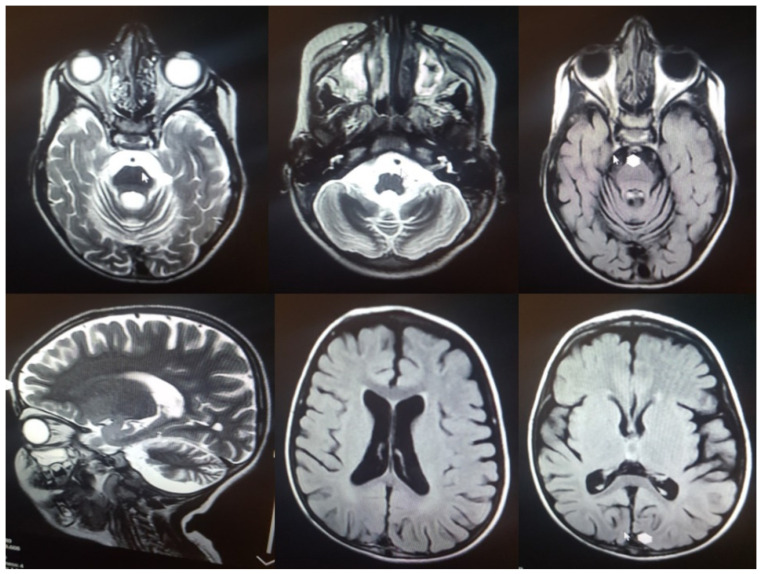
MRI findings in the index case: Hypoplasia of the cerebellar vermis and cerebellar hemispheres.

**Figure 5 jcm-11-06415-f005:**
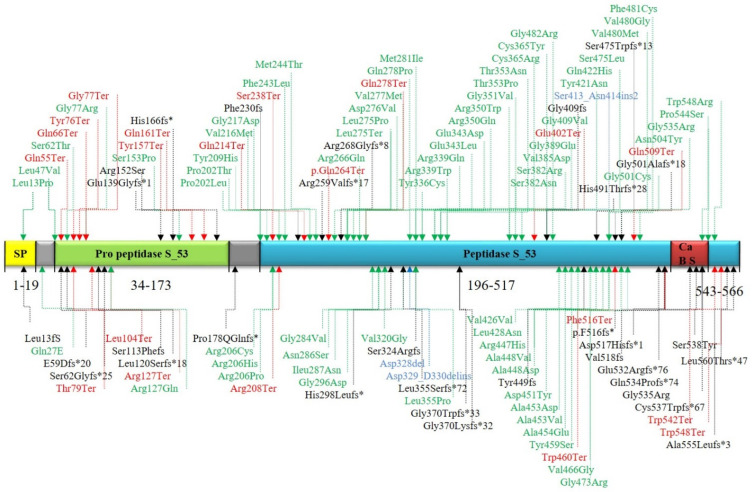
Overview of CLN2–causing TPP1 variants at protein levels. Missense mutations are marked in green while red and black indicate nonsense and frameshift mutations, respectively, and blue indicates insertions and deletions. SP: Signal peptide, Ca BS: Ca^2+^ Binding site.

**Table 1 jcm-11-06415-t001:** Pathogenic *TPP1* variants and associated clinical details (literature review).

Age/ Sex	First Symptoms/Age at Onset	Additional Symptoms	Consan-Guineous	HGVS cDNA	HGVS Protein	Exon/ Intron	ACMG Scoring	ACMG Prediction	Zygosity	Ethnicity	Reference
5y/M	Seizures/4y	Cerebellar atrophy, ataxia, epilepsy, vision loss, spasticity, tremor, motor difficulties, swallowing problems, sleep disorder, cognitive decline, language regression	yes	c.C832T	p.Q278X	7	PVS1, PM2	LP	Hom	Iran	This study
36y/M	Dysarthria, cardiac arrhythmia/4y	Severe cerebellar atrophy, dysarthria, ataxia, dystonia, right-hand tremor	No	c.38T > C c.1523A > G	p.L13P p.T508C	2 12	PM2 PM2	VUS VUS	Comp het	Poland	[32]
4y/F	Seizure/4y	Cognitive decline, motor deterioration, epilepsy, language regression, generalized hypotonia, cerebral and cerebellar atrophy	No	c.89 + 1G > A	Splice effect	2	PVS1, PM2, PP5	LP	Hom	India	[29]
9y/M	Anxiety, hypervigilance, sleep disorder/5y	Language regression, motor difficulties, microcephaly, spasticity, mild cortical and cerebellar atrophy	No	c.225A > G c.1012C > G	p.Q75Q p.Q338E	3 8	PM2, PP5 PM1, PP2	LP VUS	Comp het	Australia	[27]
6y/F	Seizures/3.5y	Mental and motor deterioration, ataxia, cerebellar atrophy, vision loss, Retinitis pigmentosa, myoclonic epilepsy	NA	c.229G > T IVS4-17~-4delTGTTCTCTGACCTC	p.G77X; Splice effect	3	PVS1, PM2	LP	Comp het	China	[30]
4.6y/M	Speech delay, mild mental retardation, autistic features/2y	Seizure, myoclonic jerks, ataxia, motor difficulties, bilateral optic atrophy	NA	c.183_184 delCT c.417G > A	p.S61fsX25 p.G473R	3 11	PVS1,PM2 PM2, PP2	LP LP	Com het	China	[33]
8y/M	Movement disorder, neuroregression/2y	Cerebellar atrophy, microcephaly	No	c.341C > T	p.A114V	4	PP2, PM2	VUS	Hom	Turkey	[7]
6y/M	Seizure/3y	Mental and motor deterioration, vision loss, ataxia, cerebellar atrophy, myoclonic epilepsy	NA	c.409-410insGCTG c.1546-1547insTTCA	p.E139G p.D517H	5 12	PM2. PP2	VUS VUS	Comp het	China	[30]
50M/F	Seizures/45M	Tremor, ataxia, cerebellar atrophy	No	c.509-1G > C	Splice effect	6	PVS1, PM2	P	Hom	Italy	[34]
-/F	Seizures/3.5y	Ventricle and axial fluid spaces, ataxia, dysarthria, dementia, language regression, vision loss, tremor	No	c.T523-1G > A	Splice effect	NA NA	PVS1, PS4	P	Hom	Scottish Irish English	[35]
20y/F	Vision failure/7y	Seizures	No	c.T523-1G > C	Splice effect	NA	PVS1, PS4	P	Hom	Lebanese	[35]
NA/F	Seizures/44M	Speech delay, regression, ataxia, myoclonus	NA	c.533del	p.P178Q	6	PVS1, PM2, PP5	P	Hom	NA	[36]
15y/M	Motor dysfunction/3y	Dementia, mental deterioration, speech difficulties, blindness, spasticity	NA	c.622C > T c.1439T > G	p.R208 X p.V480G	6 12	PVS1, PM2	P	Comp het	Czech Republic	[37]
5y/F	Seizures/ 4y	Focal abnormality, hypsarrhythmia, tremor, generalized ataxia, motor and mental regression, cerebellar atrophy, optic nerves bilateral atrophy	No	c.622C > T	p.R208X	6	PVS1, PM2	P	Hom	Swiss	[16]
5y/F	Seizures/2y	Mental and motor deterioration, ataxia, cerebellar atrophy, myoclonic epilepsy	NA	c.622C > T: c.640C > T	p.R208X; p.Q214X	6 6	PVS1, PM2; PVS1, PM2, PP5	P P	Comp het	China	[30]
12y/F	Seizures/2y	Cerebellar atrophy, ataxia, cognitive deterioration, myoclonic epilepsy, blindness, language delay	No	c.622C > T	p.R208X	6	PVS1, PM2	P	Hom	Taiwan	[28]
NA/F	Seizures/43M	Delayed speech development, behavioral and sleep abnormalities, cerebellar atrophy	NA	c.622C > T	p.R208X	6	PVS1, PM2	P	Hom	NA	[36]
NA/M	Seizures/36M	Cerebellar and cerebral Atrophy, language delay, motor difficulty, behavioral abnormalities	NA	c.622C > T	p.R208X	6	PVS1, PM2	P	Hom	NA	[36]
NA /F	Seizures/35M	Motor and language regression, ataxia, choreoathetosis, enlargement of subarachnoidspace, cerebral and cerebellar atrophy	NA	c.622C > T	p.R208X	6	PVS1, PM2	P	Hom	NA	[36]
NA/F	Seizures/36M	Language delay, motor disturbance, cerebellar atrophy	NA	c.622C > T	p.R208X	6	PVS1, PM2	P	Hom	NA	[36]
NA/F	Seizures/38M	Hypotonia of right upperlimb, delayed speech	NA	c.622C > T	p.R208X	6	PVS1, PM2	P	Hom	NA	[36]
4y/M	Seizures/2y	Cognitive and motor deterioration, speech delay, tonic-clonic seizures, muscular hypotonia, cerebral atrophy, cerebellum hypoplasia, optic nerves damage, hand tremor, ataxia	NA	c.622C > T	p.R208X	6	PVS1, PM2	P	Hom	Russian	[14]
9y/M	Seizures/3y	Mental and motor deterioration, ataxia, cerebellar atrophy, myoclonic epilepsy	NA	c.640C > T; c.650G > T	p.Q214X; p.G217D	6 6	PVS1, PM2, PP5; PM2, PM5, PM1, PP2, PP5	P P	Comp het	China	[30]
-/M	Seizures/3.5y	Mild cerebellar atrophy	No	c.646G > A	p.V216M	6	PVS1, PM2	LP	Hom	China	[2]
8y/F	Seizures/3y	Visual abnormalities, cognitive regression, mild atrophy of the cerebellum	No	c.646G > A	p.V216M	6	PVS1, PM2	LP	Hom	China	[2]
40y	-/10y	Cognitive and motor dysfunction, epilepsy, seizure.	NA	IVS7-10A > G	NA	7	NA	LP	Hom	Portugal	[31]
10y/F	Seizures, vomiting/3y	Ataxia, language regression, hyperreflexia, cerebral and cerebellar atrophy	No	c.775delC	p.Arg259fs	7	PVS1, PM2	LP	Hom	Arab	[38]
-/M	Seizures/3y	Stopped walking and talking	No	c.775delC	p.Arg259fs	7	PVS1, PM2	LP	Hom	Arab	[38]
4.5/F	Seizures/2.5y	Cognitive and motor impairment, loss of ambulation, developmental delay, speech deterioration, hypertonia, hypereflexia	No	c.1016G > A	p.R339Q	8	PM2, PM5, PM1, PP2	P	Hom	India	[1]
11M/M	Hypotonic/26day	Severe cerebellar and cerebral atrophy and no newborn reflexes	No	c.1145G > A	p.S382N	9	PM2,PM5,PP3,PP2	VUS	Hom	Turkey	[3]
5y/F	Myoclonus, nystagmus/6M	Cerebral atrophy, movement disorder, epilepsy	Yes	c.1204G > T	p.E402X	10	PVS1, PM2	LP	Hom	Turkey	[7]
7y/M	Seizures/2.5y	Drop attacks, low vision, speech and cognitive degeneration	No	c.1551 + 1insTGAT	Splice effect	12	PVS1, PM2, PP5	P	Hom	China	[13]
7y/F	Seizures/3y	Ataxia, vision loss, cerebellar atrophy, brain stem atrophy, general epileptic discharges, drop attacks	No	c.1551 + 1insTGAT	Splice effect	12	PVS1, PM2, PP5	P	Hom	China	[13]
3.7/M	Seizures/8M	Frequent spike and wave, unsteady gait, myoclonic jerks,	No	c.1551 + 1insTGAT	Splice effect	12	PVS1, PM2, PP5	P	Hom	China	[13]
5y/F	Seizures/ 3.2y	Ataxia, mild brain atrophy, myoclonic seizures, cognitive decline, motor dysfunction.	NA	c.1551 + 1G > T; c.1613C > A	Splice effect; p.S538Y	12 13	PVS1, PP5, PM2 PM2, PP3, PP2	P VUS	Comp het	China	[26]

y: year, M: month, Hom: homozygote, Comp het: compound heterozygote, P: pathogenic, LP: likely pathogenic, NA: not ascertained, M: male, F:female.

## Data Availability

Sequencing data is available on personal request for reasonable purposes.

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
