# Peer review of "Identification of a TPP1 Q278X Mutation in an Iranian Patient with Neuronal Ceroid Lipofuscinosis 2: Literature Review and Mutations Update"

_jcm, 2022, doi:10.3390/jcm11216415_

Round 1

Reviewer 1 Report

Thank you for the opportunity to review the manuscript entitled: Identification of a TPP1 Q278X mutation in an Iranian patient with Neuronal Ceroid Lipofuscinosis 2: Literature review and mutations update

This is an interesting topic with a highly quality and relevancy for patients affected with this condition.

However, I would like to do the following comments:

In line 42 seems to be a misspelling in US. Please correct accordingly.

In line 43 please add a reference to support the sentence of incidence.

In line 75 ‘isfollowed’ is misspelled.

In line 83 localisation is misspelled.

In line 86 ‘marcromolcules’ is misspelled.

In line 87 Next generation sequencing (NGS) all words should be in capital letters?

In line 140, this sentence is a little confuse. Initially the MRI findings are described as” no abnormalities” and then some important abnormalities are described. Please clarify.

This paper is a “comprehensive literature review” according with the title and description section, however in the methods is not there any description about the process, findings etc., please review and made changes accordingly.   

During the discussion in line 164 the authors describing “our cases” but they are reporting one case. This is a little confuse, please clarify. Also, in line 164 there is a period that seems to be off. Please review this paragraph for clarity.

For the clinician readers, please add a sentence in the discussion about the importance of this mutation  and what would be the teaching point.

Congratulations for this interesting and thorough job  

Author Response

We thank the reviewer for the kind suggestions. We have adressed all points raised.

In line 42 seems to be a misspelling in US. Please correct accordingly. -corrected

In line 43 please add a reference to support the sentence of incidence.-we added a reference to orphanet

In line 75 ‘isfollowed’ is misspelled.-corrected

In line 83 localisation is misspelled.-corrected

In line 86 ‘marcromolcules’ is misspelled.-corrected

In line 87 Next generation sequencing (NGS) all words should be in capital letters?-corrected

In line 140, this sentence is a little confuse. Initially the MRI findings are described as” no abnormalities” and then some important abnormalities are described. Please clarify.

We changed the sentence to “MRI) findings revealed only mild supratentorial dilation…”

This paper is a “comprehensive literature review” according with the title and description section, however in the methods is not there any description about the process, findings etc., please review and made changes accordingly.   -added

During the discussion in line 164 the authors describing “our cases” but they are reporting one case. This is a little confuse, please clarify. Also, in line 164 there is a period that seems to be off. Please review this paragraph for clarity.-corrected

For the clinician readers, please add a sentence in the discussion about the importance of this mutation  and what would be the teaching point.

-we added a statement that earlier genetic diagnosis could have impacted the clinical course in our case. Our teaching point is that early genetics is crucial in the light of novel evolving therapies

Reviewer 2 Report

It is a well written case report, which includes a comprehensive summary of the actual knowledge about NCL. I’m not competent to provide a critical appraisal of the reported pieces of information regarding the CLN classification or TPP1 variants. I assume they are correct.

As a very minor issue, I wonder whether at least part of the clinical data concerning the patients should move from the results to the material and methods section.

Author Response

We thanks the reviewer for this comment but feel the case description should remain in the main part of the text and not be moved to the methods section